# COVID-19 Infections Still Occur: How Do Pregnant and Non-Pregnant Individuals Compare? A Study from the Canadian Mother–Child Initiative on Drug Safety in Pregnancy (CAMCCO)

**DOI:** 10.3390/ijerph22111756

**Published:** 2025-11-20

**Authors:** Anick Bérard, Odile Sheehy, Padma Kaul, Sherif Eltonsy, Mark Walker, Steven Hawken, Sasha Bernatsky, Michael Pugliese, Olesya Barrett, Anamaria Savu, Roxana Dragan

**Affiliations:** 1Research Center, CHU Ste-Justine, Montreal, QC H3T 1C5, Canada; 2Faculty of Pharmacy, University of Montreal, Montreal, QC H3T 1J4, Canada; 3Faculty of Medicine, Université Claude Bernard Lyon 1, 69622 Lyon, France; 4Faculty of Medicine, University of Alberta, Edmonton, AB T6G 2B7, Canadasavu@ualberta.ca (A.S.); 5Faculty of Pharmacy, University of Manitoba, Winnipeg, MB R3T 2N2, Canada; 6Faculty of Medicine, University of Ottawa, Ottawa, ON K1N 6N5, Canada; 7Institute for Clinical Evaluative Sciences, Ottawa, ON K1Y 1J8, Canada; 8Faculty of Medicine, McGill University, Montreal, QC H3A 0G4, Canada

**Keywords:** pregnancy, people of reproductive age, COVID-19, adverse pregnancy outcomes, CAMCCO

## Abstract

Over 100 million pregnant people worldwide remain at risk of COVID-19. We compared the prevalence of severe COVID-19 in pregnancy and in people of reproductive age, and the risk of adverse pregnancy/neonatal outcomes in those with/without COVID-19 during gestation. In the Canadian Mother–Child Cohort, two sub-cohorts were identified using medical services, prescription medication fillings, hospitalizations, and COVID-19 surveillance testing programs data (28 February 2020–2021). The first included all pregnant people with at least one completed trimester of pregnancy during the study period, stratified on COVID-19 status. The second included all non-pregnant people (aged 15–45) with a positive COVID-19 test during the same period. COVID-19 severity was categorized based on hospital admissions before the end of pregnancy. Associations between COVID-19 during pregnancy and adverse perinatal outcomes were quantified using log-binomial regressions. A total of 150,345 pregnant people (3464 (2.3%) had COVID-19), and 112,073 non-pregnant people with COVID-19 were included. Maternal age at the time of COVID-19 diagnosis/positive test was statistically significantly lower among pregnant individuals compared to those who were not pregnant (96% had less than 40 years vs. 80%, *p* < 0.001). In pregnancy, COVID-19 was associated with the risk of spontaneous abortions (adjRR 1.76, 95%CI 1.37, 2.25), gestational diabetes (adjRR 1.52, 95%CI 1.18, 1.97), prematurity (adjRR 1.30, 95%CI 1.01, 1.67), and NICU (adjRR 1.32, 95%CI 1.10, 1.59); COVID-19 treatment with medications reduced risks. Severe COVID-19 was more prevalent in pregnancy and was associated with higher risks of adverse maternal/neonatal outcomes. As some countries are pulling back preventive strategies for COVID-19, this study highlights the importance of continued surveillance during pregnancy to prevent adverse pregnancy outcomes.

## 1. Introduction

COVID-19 usually presents with fever and cough, but pneumonia is frequently seen among infected patients [1]. Pregnancy is associated with a number of physiological changes that make pregnant persons more vulnerable to respiratory pathogens and severe pneumonia, including COVID-19 [2]. A systematic review showed a doubling of the risk of intensive care unit (ICU) admissions and invasive ventilation among pregnant persons with COVID-19 compared to non-pregnant persons with COVID-19 [3]. The Canadian surveillance program of COVID-19 in pregnancy (CANCOVID-preg) identified 8786 COVID-19-infected pregnancies in Canada between 1 March 2020 and 31 October 2021. Although precise estimates are difficult to ascertain in surveillance systems and head-to-head comparisons in well-defined cohorts are necessary, CANCOVID-preg reported that pregnant individuals were more likely to be hospitalized than infected non-pregnant persons of childbearing age (relative risk (RR) 2.65; 95%CI 2.41–2.88). CANCOVID-preg also estimated that COVID-19 among pregnant persons, regardless of the severity of the infection, was associated with a 63% increased likelihood of preterm birth compared to pregnant individuals without COVID-19 infection [4].

Treatment guidelines for care of patients with COVID-19 were updated regularly by the US National Institutes of Health [5]; the last update was in May 2025 [6], which included data on COVID-19-specific medications. Pregnant persons have been excluded from many clinical trials evaluating new treatments for COVID-19; however, treatments recommended for the non-pregnant population can benefit pregnant persons [7]. Given the potential higher severity of diseases during pregnancy and thus higher adverse maternal, fetal, and neonatal outcomes, it is therefore recommended that pregnant persons infected with COVID-19 should be given priority for early treatment [8]; pregnant people are also prioritized for COVID-19 vaccination for the prevention of COVID-19 during gestation [9]. In a descriptive study in Spain, Mota-Pérez et al. showed that individuals who had COVID-19 during the third trimester were using more medications, mostly antithrombotic medications of the heparin group, followed by analgesics, antibiotics, NSAIDs, and corticosteroids [10]. A significant lack of information remains about medicines used for the treatment of COVID-19 in the pregnant population, by trimester and severity of infection. Few studies have assessed the risk of pregnancy and neonatal adverse outcomes associated with prenatal use of COVID-19 indication-specific or repurposed medications. Although COVID-19-specific medications are available now, few are regularly used to treat COVID-19 during pregnancy. Hence, reliance on repurposed medications remains the norm. The health care system and access to health care are determining factors modulating use and risks, which could result in different findings between the US and Canada.

Given that there are limited data on how Canadian pregnant individuals with COVID-19 compare to non-pregnant individuals with COVID-19 in terms of disease severity and medication use as well as on head-to-head comparisons of diseased and non-diseased pregnant persons with regard to adverse maternal, pregnancy, and neonatal outcomes, we aimed to (1) compare COVID-19 severity and medication use prevalence between pregnant and non-pregnant individuals with COVID-19 and (2) describe and assess the impact of COVID-19 and medication use during pregnancy on pregnancy and neonatal outcomes using a large cohort in Canada. As global COVID-19 vaccination rates decline, it is increasingly important to revisit how pregnant individuals are affected by the virus.

Given that many countries are scaling back with all COVID-19 prevention initiatives, including vaccination, and that no inclusive pregnancy, maternal, and child data are available in Canada, our study aims to provide robust baseline estimates for the Canadian pregnant population. This is needed to assess the long-term impacts of COVID-19 in pregnancy.

## 2. Materials and Methods

**Data Sources**: The Canadian Mother–Child Cohort (CAMCCO) [11] provincial data infrastructure was used for this study. Leveraging provincial health (outpatient and physician, and emergency department visits; and hospitalizations) and sociodemographic databases, CAMCCO has put in place harmonized provincial mother–child cohorts with longitudinal follow-up of mothers and children from 1995 to 2022. CAMCCO is described in Bérard et al. [11]. Briefly, CAMCCO has developed standardized and harmonized diagnosis and medication codes, programming and SAS algorithms, common data model (which enables the capture of similar datasets with the same variables across provinces), enabling database linkages, follow-up, and identification of variables (beginning and end of pregnancy, trimester definition, medication exposure algorithms, mother–child link, major malformations, prematurity, low birth weight (LBW), etc.) in a similar manner across provinces (www.motherchildcohort.ca, accessed 18 November 2025). Given the nature of the data, crude health data cannot leave the province, and thus, provincial harmonized CAMCCO cohorts were put in place. Given the nature of the data that we have, no losses-to-follow-up are observed unless women moved out of the province, which does not happen during any studied pregnancies, or die. Given that physicians, pharmacists, and hospitals are paid for each service or medication dispensed, missing data occurs seldomly. At present, CAMCCO includes data on approximately 4 million Canadian pregnancies and 3 million children with up to 30 years of follow-up (1995–2024); CAMCCO is one of the largest and most representative longitudinal cohorts of pregnancies, mothers, and children in the world.

**Study Design**: The study is a multi-database, dynamic multiple-cohort study, conducted between 28 February 2020 and the end of 2021, covering the first four COVID-19 pandemic waves. This work was performed in collaboration with the ‘COVID-19 infectiOn aNd medicineS In pregnaNcy (CONSIGN)’ international consortium.

**Study population**: For this study, CAMCCO data from Alberta, Manitoba, and Ontario were used. In each province, two key cohorts were identified using prospective data collection of medical services, prescription medications, hospitalization archives data, and COVID-19 diagnosis and testing program results. The medical services included detailed information on all medical services, including physician-based diagnosis (*International Classification of Disease*, 9th revision (ICD-9), and 10th revision (ICD-10)), and the health care provider’s unique identifier and his/her specialty, medical procedures, as well as the calendar date when the medical procedures were performed. The prescription medication datasets cover information on all filled prescribed medications, the prescribing physician, and the pharmacist’s unique identifier, start date, drug identification number, dosage, quantity, and duration of the prescription filled. The hospitalization archives data provided information on diagnostic codes (ICD-9 and ICD-10), interventions, procedures, and consultations. Finally, COVID-19 positive tests were obtained from each province’s surveillance system. Through a unique patient-encrypted identifier, data within each province were linked together.

**Cohort 1**: Pregnant person: The first cohort included all pregnant people (aged 15–45 years) with at least one completed trimester of pregnancy during the study period (28 February 2020–31 December 2021), regardless of pregnancy status (delivery, induced/planned abortion, or spontaneous abortion). The index date was the first day of gestation (1DG), defined as the first day of the last menstrual period. In cohort 1, the analyses were conducted overall during pregnancy and by trimester. Trimesters were defined as the 1DG to day 98 after 1DG (trimester 1); from day 99 after 1DG to day 182 after 1DG (trimester 2); from day 183 after 1DG until the end of pregnancy (trimester 3). Only the first pregnancy per person after 28 February 2020 was included in the analyses. Cohort 1 was further stratified on COVID-19 status (yes/no) during pregnancy (Figure 1).

**Cohort 2**: COVID-19 positive non-pregnant women: The second cohort included all women (aged 15–45) without a pregnancy during the study period (28 February 2020–31 December 2021) with a COVID-19 diagnosis or a positive COVID-19 test. For Cohort 2, the index date was the date of the COVID-19 diagnosis or positive COVID-19 test.

Pregnant and non-pregnant people with less than one year of continuous data prior to the index date were excluded. This was performed to have sufficient time to assess covariates and potential confounders.

**COVID-19 infection**: COVID-19 status in pregnant or non-pregnant people was assessed using COVID-19 test results or ICD-10 CM code U07.1 from hospital data. The COVID-19 diagnoses or positive test results among pregnant people had to be between 1DG (index date) and the end of pregnancy.

Severity of the COVID-19 infection: COVID-19 severity was categorized into two levels: non-severe and severe, based on whether the woman was admitted to the hospital regardless of the reason, according to the World Health Organization (WHO) definition [12].

**Prevalence of medicine use**: The prevalence of medicine use was assessed using the Anatomical Therapeutic Chemical (ATC-level 2) classification system. The ATC codes for medicines of interest for COVID-19 were those available on the WHO Collaborating Centre website (17 March 2022 version). Women with COVID-19 were considered exposed to an ATC-level 2 medicine class if they had filled at least one prescription for a medicine included in the WHO list (see Appendix A) during the 30 days post-COVID-19 positive test or diagnosis; this prevalence was compared to the medicines used in the 30 days pre-COVID-19 diagnosis or positive test. The prevalence numerator was the number of women having one or more prescriptions with a date of dispensing within 30 days pre-COVID-19 or 30 days post-COVID-19 diagnosis or positive test results. The denominator was the number of women contributing at least one day of person-time to the study period. Medications, identified by outpatient prescription dispensing codes, were defined by therapeutic classes (ATC-level 2). For each province, the prevalence of medicine exposures was presented for pregnant people with COVID-19 and for non-pregnant women with COVID-19.

**Maternal, pregnancy, and neonatal outcomes**: The impact of COVID-19 during pregnancy on maternal, pregnancy, and neonatal outcomes was compared between pregnant people with and without COVID-19 overall, and stratified by pregnancy trimesters. Appendix A presents the list of outcomes with their definition. The number of women included in the analyses on maternal, pregnancy, and neonatal outcomes differed by outcome, given that spontaneous and planned abortions, as well as available trimesters (i.e., premature pregnancies have shorter pregnancy follow-up), had to be taken into account. All pregnant people are included in the analyses on spontaneous abortion and termination of pregnancy due to fetal anomaly (TOPFA). For gestational diabetes and preeclampsia, pregnant people included in the analyses had to have pregnancy durations of more than 20 weeks’ gestation. Analyses for cesarean, prematurity, and stillbirths included only people who had a delivery (live or stillbirth). Finally, for the analysis on low birth weight (LBW), small-for-gestational age (SGA), neonatal intensive care unit (NICU), and major congenital malformation (MCM), we only included people with livebirth deliveries. See Appendix A for codes used.

**Covariates**: The covariates used for the adjusted models are risk factors for the outcomes studied to answer objective 3. They included (1) maternal age; (2) maternal comorbidities associated with higher risk of COVID-19 severity measured in the year prior to the 1DG (hypertension, respiratory disease, diabetes type 1 or 2, obesity, chronic kidney disease, HIV, use of immunosuppressants, cancer, mood and anxiety disorders, other mental disorder, common rheumatic diseases); (3) previous pregnancies with history of gestational diabetes or hypertensive disorders; prior delivery with SGA newborns or newborns with major congenital malformation measured in the 2-years prior to the 1DG. All these covariates were identified from either diagnoses or disease-specific medications available in the cohorts (Appendix A).

**Data Harmonization**: To harmonize variable definitions across the provincial cohorts, exposures, covariates, and outcomes across participating provinces, we used a common protocol. We shared with the three provinces the shell tables, the code list of outcomes and exposures, as well as covariates, definitions of risk time-windows, trimester calculations, and other relevant algorithms.

**Statistical analyses**: In each cohort of pregnant persons, demographic characteristics, proxies for lifestyle (smoking and alcohol consumption), pregnancy outcomes, and maternal comorbidities were compared between people with and without COVID-19 during pregnancy. Similar comparisons were also made between COVID-19-infected pregnant people and non-pregnant women with COVID-19. Chi-Square test or Fisher’s exact test, when the expected values in any of the cells of a contingency table are below five (5), were used to compare categorical variables, and Student’s T-test in order to compare continuous variables. Prevalence of ATC level 2 medicines use during the 30 days pre- and post-COVID-19 overall and by infection severity was evaluated among pregnant people with COVID-19 and non-pregnant women with COVID-19.

Associations between COVID-19 infections during pregnancy overall and by trimester of pregnancy and outcomes of interest were quantified using log-binomial regression models, using the link function logit to calculate the relative risks (RR). Crude and adjusted estimates were obtained with 95% confidence intervals (CI). Multivariate model was adjusted for maternal age, maternal comorbidities in the year prior to the 1DG (11 independent variables considered dichotomously and independently: hypertension, respiratory disease, diabetes type 1 or 2, obesity, chronic kidney disease, HIV, use of immunosuppressant, cancer, mood and anxiety disorders, other mental disorder, common rheumatic diseases), and in the 2-years prior to the 1DG (prior history of gestational diabetes, prior history of hypertensive disorders, prior SGA newborns, prior infants with major congenital malformation). We considered all analyses significant at a *p*-value < 0.05 (two-tailed). We conducted all analyses using SAS version 9.4 software (SAS Institute Inc., Cary, NC, USA).

**Ethics**: All data used were anonymized, and all data holders in the 3 participating provinces gave permission and consent for this study. Data remained and were analyzed on secure sites within each province.

## 3. Results

From 28 February 2020 to 31 August 2021, 150,345 Canadian pregnant individuals from Alberta, Manitoba, and Ontario met the inclusion criteria, out of which 3464 had a COVID-19 diagnosis (2.3%); 112,033 non-pregnant women of reproductive age with a COVID-19 diagnosis were also included and considered for comparative purposes. Figure 2 summarizes the selection of pregnant and non-pregnant individuals studied in the three participating provinces, stratified by COVID-19 and COVID-19 severity status. Severe COVID-19 requiring hospitalization was higher in pregnant individuals (n, 393 (11.30%)) compared to non-pregnant people (n, 1737 (1.46%)) (*p* < 0.01), and this was more marked in Manitoba (Figure 2). There was a higher number of COVID-19 diagnoses during the 3rd trimester of pregnancy in the three provinces, which suggests that systematic testing was performed before delivery (Figure 2).

### 3.1. Comparisons Between Pregnant and Non-Pregnant Individuals with COVID-19

Pregnant individuals with COVID-19 were compared to non-pregnant individuals of reproductive age with COVID-19 in the three participating Canadian provinces (Table 1). Maternal age at the time of COVID-19 diagnosis/positive test was statistically significantly lower among pregnant individuals (*p* < 0.001), with more than 96% of them aged less than 40 years, as compared to 80–82% among non-pregnant people. In Alberta, 14.3% of pregnant individuals with COVID-19 had severe infections compared to 2.0% in non-pregnant people (*p* < 0.001); in Manitoba, the observed risk of severe COVID-19 among pregnant individuals was 27.7% compared to 2.6% among non-pregnant people (*p* < 0.001). No severe cases of COVID-19 infection were observed in Ontario among pregnant people. In Alberta, non-pregnant individuals with COVID-19 were more likely to have hypertension (*p* < 0.001), Type 1 and 2 diabetes (*p* = 0.02), mental disorders (*p* < 0.001), and rheumatic diseases (*p* < 0.001), but less likely to have cancer (*p* < 0.001) compared to pregnant individuals with COVID-19. In Manitoba, no differences in comorbidity profiles were observed between pregnant and non-pregnant individuals with COVID-19. In Ontario, non-pregnant women with COVID-19 were more likely to be obese (*p* = 0.009) compared to pregnant people with COVID-19; pregnant individuals with COVID-19 were more likely to have mental disorders (*p* < 0.001). Pregnant individuals with COVID-19 were more likely to have been vaccinated against influenza compared to non-pregnant people with COVID-19 (*p* = 0.41 in Manitoba, and *p* < 0.001 in Ontario).

Exposure to medications in the 30 days pre- and 30 days post-COVID-19 diagnosis was compared between pregnant and non-pregnant individuals (Figure 3). In Alberta, non-pregnant people with COVID-19 were more likely to use antihypertensives, analgesics, psycholeptics, psychoanaleptics, and medicines for obstructive airway diseases in the 30 days pre- and post-COVID-19 diagnosis compared to pregnant individuals (all *p*-values < 0.039). They were also more likely to be exposed to corticosteroids and anti-inflammatory drugs in the 30 days post COVID-19 compared to pregnant individuals with COVID-19 (*p* = 0.016 and *p* < 0.0001, respectively). On the contrary, pregnant individuals with COVID-19 were using more antithrombotic agents in the 30 days pre- and post-COVID-19 (*p* < 0.0001), and were more likely to be vaccinated against influenza in the 30 days pre-COVID-19 diagnosis (*p* < 0.0001) as compared to non-pregnant people with COVID-19. In Manitoba, a higher prevalence of anti-diabetics use in pregnant people with COVID-19 compared to non-pregnant people with COVID-19 was observed in the 30 days pre-COVID-19 diagnosis (*p* = 0.02); pregnant individuals with COVID-19 were also more likely to be exposed to anti-inflammatory drugs in the 30 days post-COVID-19 diagnosis compared to non-pregnant individuals with COVID-19 (*p* = 0.0002). However, non-pregnant individuals with COVID-19 had higher use of psychoanaleptics in the 30 days pre- and post-COVID-19 diagnosis, and higher exposure to medicines for obstructive airway disease in the 30 days post-COVID-19, compared to pregnant individuals with COVID-19 (all *p*-values < 0.04). Finally, in Ontario, non-pregnant individuals were more likely vaccinated against influenza in the 30 days post COVID-19 diagnosis, and used more psychoanaleptics in the 30 days pre- and post-COVID-19 compared to pregnant individuals with COVID-19 (all *p*-values ≤ 0.005). In all provinces, we observed a larger increase in medication use in the 30 days post-COVID-19 diagnosis compared to the 30 days prior to the COVID-19 diagnosis in individuals with severe infections as compared to those with non-severe infections, in both pregnant and non-pregnant individuals. Additional results are available in Appendix A.

### 3.2. Pregnant Individuals: Comparisons Between Those with and Without COVID-19

Pregnant individuals with and without COVID-19 were similar with regard to maternal age at the beginning of gestation, previous pregnancy history (children with major malformations, small for gestational age, stillbirths), lifestyles (tobacco or alcohol dependence) (Table 2). In the three participating CAMCCO provinces, gestational age at delivery was higher among pregnant individuals with COVID-19 compared to those without COVID-19 (*p* < 0.01) (Table 2). However, the mean gestational age at the end of pregnancy was less than 37 weeks’ gestation (prematurity) in both those with and without COVID-19 in Manitoba and Ontario (Table 2), which could be explained by the differences in sociodemographic characteristics of those covered by provincial medication insurance programs across participating provinces. Vaginal deliveries compared to cesarean section deliveries were only lower in those with COVID-19 compared to those without in Alberta (66.2% vs. 68.4%) (*p* < 0.05); no differences were found in Manitoba or Ontario (Table 2). The number of spontaneous abortions in the three provinces was lower among women with COVID-19 compared to those without COVID-19. Regarding maternal co-morbidities in the year prior to pregnancy (1DG), women with COVID-19 were generally similar to pregnant women without COVID-19 (Table 1). However, chronic diabetes and mental disorders (mood and anxiety disorders and other mental disorders) were statistically significantly higher among pregnant individuals with COVID-19 in Ontario (*p* < 0.01) compared to pregnant individuals without COVID-19 (Table 2). The history of gestational diabetes in the 2 years prior to pregnancy among pregnant people with COVID-19 was higher compared to those without COVID-19 in Manitoba (*p* = 0.001) (Table 2). Influenza vaccination was statistically significantly lower among those with COVID-19 compared to those without COVID-19 (Table 2).

The risk of adverse maternal, pregnancy, and neonatal outcomes associated with COVID-19 during pregnancy and adjusted for potential confounders across Alberta, Manitoba, and Ontario, and stratified by trimester of diagnosis, is presented in Figure 4. Adjusting for potential confounders and trimester of diagnosis, COVID-19 during pregnancy, especially during the first trimester, was associated with an increased risk of spontaneous abortion (*p* < 0.05) and gestational diabetes (*p* < 0.05). A signal for an increased risk of prematurity and small for gestational age newborn associated with COVID-19 in the third trimester of pregnancy was observed in Manitoba only (RR 1.80, 95% CI 1.27, 2.56; RR 2.28, 95%CI 1.20, 4.34, respectively). The risk for NICU admission was increased with COVID-19 in the third trimester of pregnancy (*p* < 0.05). A signal for major malformations associated with COVID-19 during the first trimester of pregnancy was only observed in Ontario (RR 2.08, 95%CI 1.18–3.69) (Figure 4).

Stratifying on the severity of COVID-19, COVID-19 was consistently associated with statistically significantly increased risk of all studied maternal, pregnancy, and neonatal outcomes based on unadjusted estimates due to small sample sizes. These are provided as signal indicators.

## 4. Discussion

This study is the first in Canada to be performed in three Canadian provinces and comparing prevalence, severity, and use of repurposed medications between pregnant people with and without COVID-19 and non-pregnant women with COVID-19. Prevalence of COVID-19 during pregnancy varied between the three included provinces, with Alberta having the highest prevalence and Ontario the lowest, which coincided with the level of restrictions imposed to control the spread of COVID-19 between 2020 and 2021 (Alberta having the least and Ontario having the most restrictions). Of the more than 262,000 people included in this study, pregnant people were more likely to have severe infections compared to non-pregnant people with COVID-19 (11.40% vs. 1.60%, *p* < 0.001). In pregnant people, Manitoba had by far the highest prevalence of severe COVID-19 among the three studied provinces. The most frequent medications used in pregnancy to treat COVID-19 were antibacterials (13.96%), psychoanaleptics (7.35%), and medicines for obstructive airway disease (3.20%). Pregnant people with a COVID-19 infection in the three participating provinces were younger than non-pregnant women with COVID-19 infections and had fewer comorbid conditions in the year prior to the index date. In pregnancy, COVID-19 was associated with a higher risk of spontaneous abortions (adjRR 1.76, 95% CI 1.37, 2.25), gestational diabetes (adjRR 1.52, 95% CI 1.18, 1.97), prematurity (adjRR 1.30, 95%CI 1.01, 1.67), and NICU admissions (adjRR 1.32, 95% CI 1.10, 1.59); COVID-19 treatment with study medications reduced all risks. The impact of COVID-19 during pregnancy on maternal, pregnancy, and neonatal outcomes varied according to the timing of the COVID-19 infection during pregnancy and the province. The most consistent associations across provinces were between COVID-19 infection during the 3rd trimester and the risk of NICU, with a 31% increased risk in Alberta and 64% in Manitoba. Severe COVID-19 infection was associated with a higher risk of maternal, pregnancy, and neonatal outcomes as compared to non-severe COVID-19 infection among pregnant women.

Our findings are consistent with those of CANCOVID-preg [4]. Although CANCOVID-preg is a surveillance system with inaccurate denominators, it remains that our prevalences come from Canadian provinces that were included in CANCOVID-preg. However, our findings differ from Shinde et al. [11], who showed that pregnant people included in their study were more likely to have less severe COVID-19 than non-pregnant people of reproductive age. This could partly be explained by the fact that Schinde et al. [13] used data from workers with private health insurance in the US, hence of higher socio-demographic (SES) level and less comorbidities on average than in the three participating CAMCCO provinces, which included everyone regardless of SES status.

In the US, McClymont et al. [14] showed that COVID-19 infection among pregnant people, regardless of the severity, was associated with a 63% increased risk of prematurity compared to pregnant people without COVID-19 [14]. Our study showed a 30% increase in the risk of prematurity associated with COVID-19 in pregnancy. Although COVID-19 during pregnancy increased the risk of prematurity in both Canada and the US during the same time period, the extent of the increase was less marked in Canada. This could partly be explained by the differences in pandemic-related restrictions, socio-demographic statuses, COVID-19 severity, and health care system access between Canada and the US [15]. Similarly to our findings, Allotey J et al. [16] showed that neonates born to pregnant people infected with COVID-19 were more likely to have NICU admission (OR 4.89, 95% CI 1.87 to 12.81; 10 studies, 5873 neonates) compared to those born to pregnant people without COVID-19. Although variations between CAMCCO provinces were seen, the increase in the risk of NICU admissions associated with COVID-19 during pregnancy remained much lower in Canada than in the US, a 31% increase in Alberta and 64% increase in Manitoba. Again, differences in COVID-19 severity and health care system access between Canada and the US could partly explain the variations [15].

Very few studies have been published on the impact of the COVID-19 pandemic and the use of medications during pregnancy. Schinde et al. [13], in the US, found that pregnant people with COVID-19 were treated with anti-bacterials, anti-inflammatories, and analgesics in the 30 days after diagnosis, which is similar to our findings on anti-bacterials, and to a smaller extent, Canadian pregnant individuals with COVID-19 also took psychoanaleptics, and medicines for obstructive airway disease.

CAMCCO [11] is a collaborative research infrastructure that uses real-world data from large provincial health care databases in Canada. This allows for analyses of large sample sizes of people of reproductive age, pregnant people, and children, using already available standardized cohorts, algorithms and definitions, and harmonized protocols across provinces. Although the majority of variables used for this study have been validated [11], it remains that it was the first time that we used diagnoses of COVID-19 with ICD-10 codes or with in-hospital testing. We have used the same methodology that has been used in other studies on COVID-19, as well as in CONSIGN, and assumed that these were valid. It remains, however, that misclassification with regard to COVID-19 occurrence cannot be ruled out. Although we did not have a clinical assessment of the severity of COVID-19, we have used the WHO COVID-19 severity scale, which requires hospitalization. Even if we did not consider the hospitalization at the time of delivery to assess severity, it is possible that there is residual misclassification in the assessment of this variable, which could partly explain the very low prevalence of severe COVID-19 cases in Ontario. Furthermore, while we used the WHO definition to define COVID-19 severity, and this is a common and pragmatic approach, it remains that it could be subject to reverse causality in some cases when COVID-19 is diagnosed at the time of hospitalization, even when excluding delivery. Furthermore, our definition of COVID-19 severity could also be susceptible to ascertainment bias when comparing pregnant and non-pregnant populations, due to the fact that the threshold for hospital admission is likely different between these two groups. Pregnant individuals are, rightly, monitored more closely by the health care system. Both patients and clinicians may have a lower threshold for seeking care or recommending hospitalization for respiratory symptoms, out of an abundance of caution for both the parent and the fetus. In contrast, a non-pregnant person of the same age with identical symptoms might be more likely to be managed as an outpatient. This differential surveillance likely inflates the prevalence of severe COVID-19 in the pregnant cohort compared to the non-pregnant cohort. Our data on medications is based on prescription fillings and not actual intake. Zhao et al. [17] have shown that pregnant individuals who fill a prescription take at least one pill; hence, dichotomizing medication exposure is somewhat reflective of intake. If anything, it leads to an underestimation of the actual intake. Our study included data from Alberta, Manitoba, and Ontario; it was not possible to access the data from the other provinces in a timely and expedited manner. Nevertheless, this gave the opportunity to assess the impact of varying pandemic-related restrictions across these three provinces and should not limit generalizability.

## 5. Conclusions

Our study showed that the severity of COVID-19 was greater in pregnancy, and prevalence varied by provinces. This coincided with the level of restrictions imposed on the general population during the first four COVID-19 waves in 2020–2021. As some countries are pulling back preventive strategies for COVID-19, including vaccinations, our study suggests that surveillance and treatment of COVID-19 should not be interrupted based on pregnancy status in order to prevent severe complications of the infection. Our study provides the first all-inclusive COVID-19 pregnancy, maternal, and child data in Canada, which gives robust baseline estimates that can be used to assess the long-term impact of COVID-19 in pregnancy. These findings are extremely relevant to patient care. Considering that the COVID-19 vaccine is recommended during pregnancy by the Society of Obstetricians and Gynecologists of Canada, and that vaccination for COVID-19 decreases the prevalence and the severity of the infection [18], it is essential to find ways to encourage the general pregnant population to vaccinate even when the height of the pandemic seems to have passed.

## Figures and Tables

**Figure 1 ijerph-22-01756-f001:**
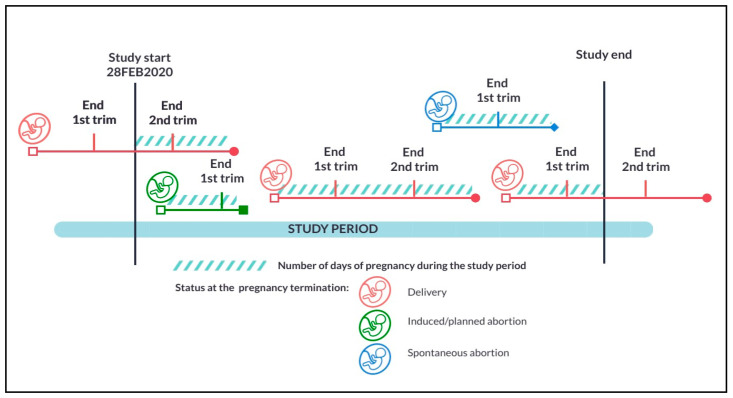
Description of Cohort 1.

**Figure 2 ijerph-22-01756-f002:**
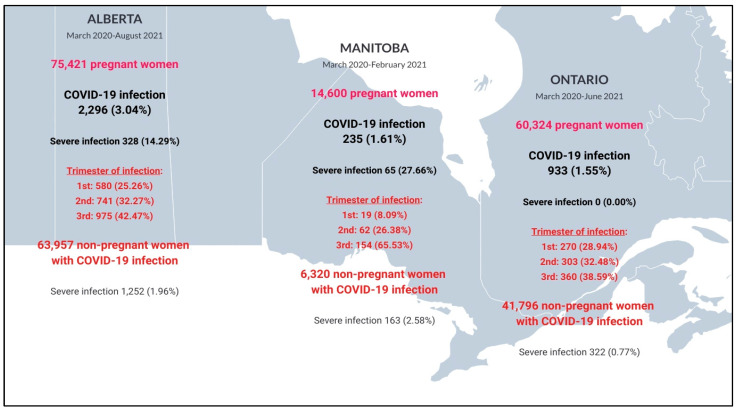
Study cohort presentation with prevalence of COVID-19 diagnoses/positive tests in Alberta, Manitoba, and Ontario.

**Figure 3 ijerph-22-01756-f003:**
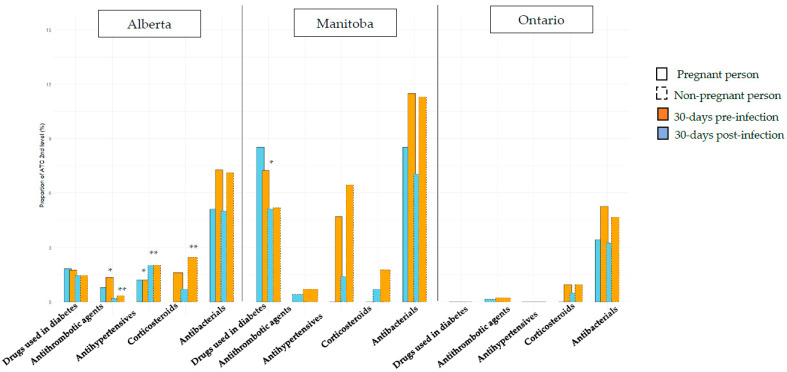
Prevalence of medication prescribed/dispensed in the 30 days pre- and post-COVID-19 positive test/diagnosis among pregnant and non-pregnant women, in Alberta, Manitoba, and Ontario. * *p* < 0.05 when comparison between pregnant and non-pregnant persons for 30 days pre-infection. ** *p* < 0.05 when comparison between pregnant and non-pregnant persons for 30 days post-infection.

**Figure 4 ijerph-22-01756-f004:**
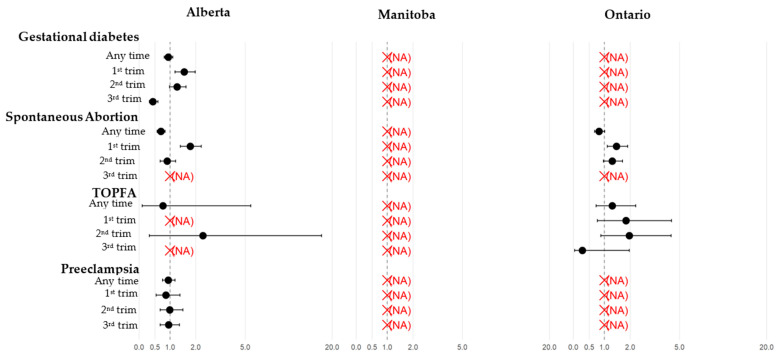
Association between COVID-19 infection and maternal, pregnancy, and neonatal outcomes during pregnancy and by trimester in Alberta, Manitoba, and Ontario. Adjusted for maternal age, maternal comorbidities in the year prior to the 1DG (11 independent variables considered dichotomously and independently: hypertension, respiratory disease, diabetes type 1 or 2, obesity, chronic kidney disease, HIV, use of immunosuppressant, cancer, mood and anxiety disorders, other mental disorder, common rheumatic diseases) and in the 2 years prior to the 1DG (prior history of gestational diabetes, prior history of hypertensive disorders, prior SGA newborns, prior infants with major congenital malformation). NA, not applicable because we have no outcome data, or the cell sizes are small and thus no access to them.

**Table 1 ijerph-22-01756-t001:** Baseline characteristics of pregnant and non-pregnant individuals with COVID-19 in Alberta, Manitoba, and Ontario.

	Alberta			Manitoba			Ontario		
Characteristics	Pregnant People with COVID-19 Infection n = 2296	Non-Pregnant People with COVID-19 Infection n = 63,957	*p*-Value	Pregnant People with COVID-19 Infection n = 235	Non-Pregnant People with COVID-19 Infection n = 6320	*p*-Value	Pregnant People with COVID-19 Infection n = 933 (1.55%)	Non-Pregnant People with COVID-19 Infection n = 41,796	*p*-Value
**Follow-up in months after index date —mean ± SD**	8.5 ± 1.1	6.2 ± 3.2	<0.001	7.7 ± 1.5	3.7 ± 1.6	<0.001	7.5 ± 2.1	6.2 ± 3.2	<0.001
**Maternal age at the 1st day of gestation (1DG) mean ± SD**	30.0 ± 5.2	30.4 ± 8.9	<0.001	28.5 ± 6.1	30.0 ± 8.8	0.045	28.1 ± 6.2	28.7 ± 9.0	0.045
**Severe COVID-19 infection**	328 (14.3)	1252 (2.0)	<0.001	65 (27.7)	163 (2.6)	<0.001	0 (0.0)	322 (0.7)	N.A.
**Site of the COVID-19 diagnosis—n (%)**			<0.001			<0.001			<0.001
Acute inpatient hospitalization	30 (1.3)	398 (0.6)		65 (27.7)	163 (2.6)		70 (7.5)	361 (0.9)	
Physician’s office/Test site	2266 (98.7)	63,559 (99.4)		170 (72.3)	6157 (97.4)		863 (92.5)	41,435 (99.1)	
**Maternal morbidity and vaccination during pregnancy or during study period for the non- pregnant women:**									
Influenza vaccination				68 (28.9)	1465 (23.2)	0.0407	115 (12.3)	805 (1.9)	<0.001
Pneumococcal vaccination				0 (0.0)	18 (0.3)	N.A.	0 (0.0)	0 (0.0)	N.A.
Pertussis vaccination				93 (39.6)	305 (4.8)	<0.0001	0 (0.0)	0 (0.0)	N.A.
**Proxy for maternal lifestyle in the year prior to the index date:**									
Tobacco dependence	27 (1.2)	873 (1.4)	0.442	8 (3.4)	168 (2.7)	0.4872	<6	67 (0.2)	N.A.
Alcohol dependence	35 (1.5)	850 (1.6)	0.868	10 (4.3)	275 (4.4)	0.9435	<6	135 (0.3)	N.A.
**Co-morbid conditions in the year prior to the index date1:**									
Cardiovascular disease	58 (2.5)	1530 (2.4)	0.68	<6	95 (1.5)	0.5856	0 (0.0)	0.316	
Hypertension	92 (4.0)	4467 (7.0)	<0.001	16 (6.8)	570 (9.0)	0.2435	14 (1.5)	0.193	
Sickle cell disease	0 (0.0)	22 (0.03)	0.37	0 (0.0)	<6	N.A.	<6	N.A.	
Respiratory disease	175 (7.6)	5769 (9.0)	0.02	17 (7.2)	503 (8.0)	0.6864	24 (2.6)	0.523	
Diabetes type 1 or 2	78 (3.4)	2456 (3.8)	0.28	11 (4.7)	472 (7.5)	0.1083	34 (3.6)	0.728	
Obesity	41 (1.8)	1335 (2.1)	0.32	8 (3.4)	109 (1.7)	0.0718	11 (1.2)	0.009	
Chronic kidney disease	29 (1.3)	1084 (1.7)	0.11	<6	81 (1.3)	N.A.	<6	N.A.	
HIV	<10	113 (0.2)	0.13	9 (3.8)	288 (4.6)	0.5987	<6	N.A.	
Use of immunosuppressants	107 (4.7)	3087 (4.8)	0.71	10 (4.3)	239 (3.8)	0.7092	15 (1.6)	0.102	
Cancer	241 (10.5)	5541 (8.7)	<0.001	<6	94 (1.5)	0.7803	45 (4.8)	0.241	
Any mental disorders	512 (22.3)	19,485 (30.5)	<0.001	51 (21.7)	1659 (26.3)	0.119	150 (16.1)	<0.001	
Mood and anxiety disorders	447 (19.5)	17,197 (26.9)	<0.001	49 (20.9)	1542 (24.4)	0.2129	131 (14.0)	<0.001	
Other mental disorders	193 (8.4)	8156 (12.8)	<0.001	12 (5.1)	458 (7.3)	0.2117	32 (3.4)	0.793	
Common rheumatic diseases	70 (3.1)	3110 (4.9)	<0.001	<6	233 (3.7)	0.0082	36 (3.9)	0.262	
Any previous co-morbid conditions	962 (41.9)	30,567 (47.8)	<0.001	91 (38.7)	2721 (43.1)	0.1878	270 (28.9)	0.115	

**Table 2 ijerph-22-01756-t002:** Baseline characteristics of pregnant individuals with and without COVID-19 in Alberta, Manitoba, and Ontario (note: NA: not applicable or no access to small cells that would allow calculations).

	Alberta			Manitoba			Ontario		
	Pregnant Women ( n = 75,421)			Pregnant Women (n = 14,600)			Pregnant Women ( n = 60,324)		
Characteristics	COVID-19 Infection n = 2296 (3.04%)	No COVID-19 Infection n = 73,125	*p*-Value	COVID-19 Infection n = 235 (1.61%)	No COVID-19 Infection n = 14,365	*p*-Value	COVID-19 Infection n = 933 (1.55%)	No COVID-19 Infection n = 59,391	*p*-Value
**Maternal age at the 1st day of gestation (1DG)**—mean ± SD	30.0 ± 5.2	30.3 ± 5.2	0.59	28.5 ± 6.1	29.1 ± 5.8	0.12	28.08 ± 6.21	28.19 ± 6.39	0.29
**Status at the end of the pregnancy**—n (%)			0.61			0.0003			<0.001
Delivery	2175 (93.5)	66,921 (90.3)		N.A.	10,594 (73.7)		604 (64.7)	32,249 (54.3)	
Planned/induced abortion	26 (1.1)	1512 (2.0)		14 (6.0)	1448 (10.1)		243 (26.0)	20,431 (34.4)	
Spontaneous abortion	125 (5.4)	5710 (7.7)		20 (8.5)	2236 (15.6)		86 (9.2)	6711(11.3)	
**Gestational age at the end of the pregnancy**—mean ± SD	37.3 ± 4.6	36.9 ± 5.5	<0.001	35.6 ± 6.6	34.1 ± 8.0	0.0004	31.8 ± 9.1	29.9 ± 9.5	<0.001
**Sex of the baby(ies)**—n (%)			0.27			0.48			0.53
Female	1044 (48.2)	32,375 (48.6)		92 (46.0)	5216 (48.5)		308 (50.2)	16,036 (48.9)	
Male	1121 (51.8)	34,240 (51.4)		108 (54.0)	5535 (51.5)		306 (49.8)	16,776 (51.1)	
**Type of delivery**—n (%)			0.03			0.68			0.37
Vaginal	1419 (66.2)	45,076 (68.4)		145 (72.1)	7822 (73.2)		482 (79.8)	25,442 (78.9)	
cesarean section	726 (33.8)	20,827 (31.6)		56 (27.9)	2859 (26.8)		122 (20.2)	6802 (21.1)	
**Proxy for maternal lifestyle in the year prior to the 1DG:**									
Tobacco dependence	27 (1.2)	1086 (1.5)	0.26	8 (3.4)	356 (2.5)	0.37	≤5	181 (0.3)	N.A.
Alcohol dependence	35 (1.5)	828 (1.1)	0.10	10 (4.3)	450 (3.1)	0.33	≤5	249 (0.4)	N.A.
**Co-morbid conditions in the year prior to the 1DG:**									
Cardiovascular disease	58 (2.5)	1964 (2.7)	0.69	<6	195 (1.4)	0.77	0 (0.0)	18 (0.0)	N.A.
Hypertension	92 (4.0)	3186 (4.4)	0.45	16 (6.8)	809 (5.6)	0.44	14 (1.5)	599 (1.0)	0.14
Sickle cell disease	0 (0.0)	19 (0.03)	N.A.	0 (0.0)	<6	N.A.	<6	73 (0.1)	N.A.
Respiratory disease	175 (7.6)	6611 (9.0)	0.02	17 (7.2)	1322 (9.2)	0.30	24 (2.6)	1628 (2.7)	0.75
Diabetes type 1 or 2	78 (3.4)	2512 (3.4)	0.97	11 (4.7)	578 (4.0)	0.61	34 (3.6)	1182 (2.0)	<0.001
Obesity	41 (1.8)	990 (1.4)	0.10	8 (3.4)	291 (2.0)	0.16	11 (1.2)	979 (1.6)	0.26
Chronic kidney disease	29 (1.3)	965 (1.3)	0.89	<6	81 (0.6)	0.15	<6	83 (0.1)	N.A.
HIV	<10	73 (0.1)	0.61	9 (3.8)	589 (4.1)	0.84	<6	86 (0.1)	N.A.
Use of immunosuppressants	107 (4.7)	3109 (4.3)	0.37	10 (4.3)	706 (2.0)	0.64	15 (1.6)	1393 (2.3)	0.14
Cancer	241 (10.5)	8147 (11.1)	0.35	<6	532 (3.7)	0.11	45 (4.8)	2959 (5.0)	0.83
Any mental disorders	512 (22.3)	17,325 (23.7)	0.13	51 (21.7)	3629 (25.3)	0.21	150 (16.1)	13,414 (22.6)	<0.01
Mood and anxiety disorders	447 (19.5)	15,010 (20.5)	0.23	49 (20.9)	3392 (23.6)	0.32	131 (14.0)	11,912 (20.1)	<0.01
Other mental disorders	193 (8.4)	6500 (8.9)	0.44	12 (5.1)	910 (6.3)	0.44	32 (3.4)	2864 (4.8)	0.05
Common rheumatic diseases	70 (3.1)	2192 (3.0)	0.94	<6	322 (2.2)	0.06	36 (3.9)	1729 (2.9)	0.09
Any previous co-morbid conditions	962 (41.9)	31,464 (43.0)	0.29	91 (38.7)	6119 (42.6)	0.23	270 (28.9)	20,073 (33.8)	0.002
**Adverse pregnancy outcome in the 2-years prior to the 1DG:**									
Prior history of gestational diabetes	75 (3.3)	2275 (3.1)	0.72	26 (11.1)	824 (5.7)	0.001	19 (2.0)	948 (1.6)	0.29
Prior history of hypertensive disorders	37 (1.6)	835 (1.1)	0.05	7 (3.0)	259 (1.8)	0.21	≤5	310 (0.5)	N.A.
Any of the above adverse reproductive history	103 (4.5)	2959 (4.1)	0.33	46 (19.6)	2115 (14.7)	0.04	23 (2.5)	1216 (2.0)	0.37
**Previous adverse reproductive history in the 2 years prior to the 1DG:**									
Prior Stillbirth	<10	247 (0.3)	0.43	<6	74 (0.5)	N.A.	0 (0.0)	9 (0.0)	0.71
Prior small for gestational age	68 (3.0)	1950 (2.7)	0.43	<6	44 (0.3)	N.A.	21 (2.3)	1372 (2.3)	0.91
Prior congenital malformation	66 (2.7)	1992 (2.7)	0.71	7 (3.0)	274 (1.9)	0.23	7 (0.8)	832 (1.4)	0.09
Any of the above adverse reproductive history	124 (5.4)	3810 (5.2)	0.72	10 (4.3)	385 (2.7)	0.14	26 (2.8)	2103 (3.5)	0.22
**Maternal morbidity and vaccination during pregnancy:**									0.03
Influenza vaccination	N.A.	N.A.		44 (18.7)	3140 (21.9)	0.009	115 (12.3)	8818 (14.8)	
Pneumococcal vaccination	N.A.	N.A.		0 (0.0)	<6	N.A.	0 (0.0)	0 (0.0)	
Pertussis vaccination	N.A.	N.A.		93 (39.6)	5844 (40.7)	0.73	0 (0.0)	0 (0.0)	

## Data Availability

Institutional contacts for data access: (1) in QC, CHU Ste-Justine Ethics Committee (http://chusj.nagano.ca, 1 May 2020), Montreal, QC, Canada for researchers who meet the criteria for access to confidential data; (2) in MB, University of Manitoba and MCHP Institutional Data Access/Ethics Committee (https://umanitoba.ca/manitoba-centre-for-health-policy/, 1 May 2020) for researchers who meet the criteria for access to confidential data; (3) in AB, University of Alberta Ethics Committee, (www.ualberta.ca/research/research-support/research-ethics-office/human-research-ethics/research-ethics-boards/, 1 May 2020).

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
