# Peer review of "COVID-19 Infections Still Occur: How Do Pregnant and Non-Pregnant Individuals Compare? A Study from the Canadian Mother–Child Initiative on Drug Safety in Pregnancy (CAMCCO)"

_ijerph, 2025, doi:10.3390/ijerph22111756_

Round 1
Reviewer 1 Report (Previous Reviewer 2)
Comments and Suggestions for Authors
Authors have revised all comments and suggestions from reviewer.
Author Response
Comment: Authors have revised all comments and suggestions from reviewer.
Answer: We thank the reviewer for their approval of our revisions.

Reviewer 2 Report (Previous Reviewer 3)
Comments and Suggestions for Authors
Dear authors,
Thank you for the revised manuscript. I've gone through it and I'm really pleased with the changes you've made. You've addressed the main concerns I had, particularly around the ascertainment bias issue and the potential for reverse causality when defining COVID-19 severity. The new paragraph in the Discussion does a great job of acknowledging these limitations transparently, which I think strengthens the paper considerably. I also appreciate that you've toned down the conclusion about severity; it reads much more carefully now and avoids overstating the findings. The plots are a nice touch too; it makes the key results much easier to digest at a glance. Overall, the manuscript is in much better shape. Well done on the revisions.
Best regards.
Author Response
Comments and Suggestions for Authors
Dear authors,
Thank you for the revised manuscript. I've gone through it and I'm really pleased with the changes you've made. You've addressed the main concerns I had, particularly around the ascertainment bias issue and the potential for reverse causality when defining COVID-19 severity. The new paragraph in the Discussion does a great job of acknowledging these limitations transparently, which I think strengthens the paper considerably. I also appreciate that you've toned down the conclusion about severity; it reads much more carefully now and avoids overstating the findings. The plots are a nice touch too; it makes the key results much easier to digest at a glance. Overall, the manuscript is in much better shape. Well done on the revisions.
Best regards.
Response: We thank the reviewer for their approval of our revisions.
Reviewer 3 Report (New Reviewer)
Comments and Suggestions for Authors
Dear Edotros,
The manuscript addressing the relationship between COVID-19 infection and adverse pregnancy and neonatal outcomes presents a well-structured analysis based on robust data, however, it appears to be of limited contemporary relevance. The central theme—examining the effects of SARS-CoV-2 during pregnancy—has been extensively explored in the scientific literature over the past several years, and the manuscript does not provide novel insights that would significantly advance the current understanding of this topic.
The reliance on data collected between 2020 and 2021 further limits the contribution of the study. The epidemiological landscape of COVID-19 has evolved substantially since that period, with widespread vaccination, viral mutation, and changing public health strategies markedly altering both disease severity and population risk profiles. Consequently, conclusions drawn from early-pandemic data are of diminished applicability to the present day.
Moreover, the manuscript largely reiterates findings that have already been well established in previous meta-analyses and large-scale cohort studies—namely, the association between COVID-19 infection in pregnancy and increased risks of spontaneous abortion, gestational diabetes, preterm birth, and neonatal intensive care admission. The lack of novel hypotheses, updated data, or extended longitudinal follow-up reduces the originality and scientific impact of the work.
From a methodological standpoint, while the statistical analyses appear appropriate, the research questions themselves are no longer aligned with current scientific priorities. Contemporary investigations have shifted toward understanding long-term maternal and neonatal consequences of COVID-19, post-infection sequelae, and the differential effects of vaccination and viral variants—none of which are addressed in this study.
In summary, although the study is competently executed, it contributes little to the advancement of current scientific discourse. The outdated focus and reliance on early-pandemic data substantially limit its relevance, rendering the manuscript more suitable as a retrospective summary than as an original research article in the current academic context.
Author Response
Comments and Suggestions 1:
Dear Edotros, The manuscript addressing the relationship between COVID-19 infection and adverse pregnancy and neonatal outcomes presents a well-structured analysis based on robust data, however, it appears to be of limited contemporary relevance. The central theme—examining the effects of SARS-CoV-2 during pregnancy—has been extensively explored in the scientific literature over the past several years, and the manuscript does not provide novel insights that would significantly advance the current understanding of this topic. The reliance on data collected between 2020 and 2021 further limits the contribution of the study. The epidemiological landscape of COVID-19 has evolved substantially since that period, with widespread vaccination, viral mutation, and changing public health strategies markedly altering both disease severity and population risk profiles. Consequently, conclusions drawn from early-pandemic data are of diminished applicability to the present day. Moreover, the manuscript largely reiterates findings that have already been well established in previous meta-analyses and large-scale cohort studies—namely, the association between COVID-19 infection in pregnancy and increased risks of spontaneous abortion, gestational diabetes, preterm birth, and neonatal intensive care admission. The lack of novel hypotheses, updated data, or extended longitudinal follow-up reduces the originality and scientific impact of the work.
Response 1: We thank the reviewer for this comment. However, we strongly disagree with the fact that widespread vaccination is present in the population and in pregnant women specifically. This is not the case in Canada and in the US – the vaccination rates are decreasing. COVID-19 fatigue is present, and thus a clear reminder of the impact of COVID-19 during pregnancy on maternal, pregnancy, and child health is still very much needed. In addition, our study is all encompassing in the way that it tackles COVID-19 severity during pregnancy and impact on the combined pregnancy, maternal and child outcomes – this has never been done. Finally, given the cultural effect modification of COVID-19, country-specific data are needed, even more so at present. No such data are available in Canada.
The following has been added on Lines 98-102:
Given that many countries are scaling back with all COVID-19 prevention initiatives, including vaccination, and that no inclusive pregnancy, maternal and child data are available in Canada, our study aims to provide robust baseline estimates for the Canadian pregnant population. This is needed to assess the long-term impacts of COVID-19 in pregnancy.
Comments 2: From a methodological standpoint, while the statistical analyses appear appropriate, the research questions themselves are no longer aligned with current scientific priorities. Contemporary investigations have shifted toward understanding long-term maternal and neonatal consequences of COVID-19, post-infection sequelae, and the differential effects of vaccination and viral variants—none of which are addressed in this study.
Response 2: We agree with the reviewer. However, before doing this, we need baseline evidence from the Canadian population. Hence the importance of our study to provide robust all-inclusive baseline estimates.
The following has been added to the Conclusion on Lines 531-534: Our study provides the first all-inclusive COVID-19 pregnancy, maternal and child data in Canada, which gives robust baseline estimates that can be used to assess the long-term impact of COVID-19 in pregnancy. These findings are extremely relevant to patient care.
This manuscript is a resubmission of an earlier submission. The following is a list of the peer review reports and author responses from that submission.
Round 1
Reviewer 1 Report
Comments and Suggestions for Authors
Congratulations for this interesting work. An important topic had been addressed in spite of the limitations mentioned about the methodology.
kindly address the following comments:
Results:
- Table1, page 357, second row (site of COVID diagnosis): please check the format and properly align the numbers with text.
- Table 2 & 3 N.A is not defined, and is written inconsistently, sometimes with small letters and sometimes capital letters , please unify them
- Table 4 can be simplified by omitting the results in each trimester but keeping the data related to anytime during pregnancy. any significant results related to any of the trimesters can mentioned by text in the result's paragraph.
Conclusion:
don't repeat the results in the conclusion, and emphasize more on the possible clinical benefits of such study.
Reviewer 2 Report
Comments and Suggestions for Authors
The research topic holds promise; however, the topic is not quite relevant with current situation as not having a pandemic. The manuscript requires significant revisions to improve the quality and interpretation of the data, enhance its scientific communication, and correct numerous language and structural issues. As it stands, I do not recommend acceptance for publication in its current form. I encourage the authors to revise the manuscript thoroughly, addressing these concerns before submitting it elsewhere.

Reviewer 3 Report
Comments and Suggestions for Authors
Dear Authors,
You have undertaken a study on a topic of great importance. Understanding the specific risks posed by COVID-19 to pregnant individuals and their newborns is a public health priority, and your work provides valuable epidemiological insights from a large, population-based Canadian Cohort.
The resume effectively frames the research question, and the use of the CAMCCO platform to analyse over 150,000 pregnancies lends significant statistical power to your findings on perinatal outcomes. The associations you identified between COVID-19 infection and increased risks of spontaneous abortion, gestational diabetes, and prematurity are clear, well-quantified, and represent a significant contribution to the field. My primary concern relates to the methodology used to compare the severity of COVID-19 between pregnant and non-pregnant cohorts.
Your study defines COVID-19 severity based on hospital admissions. While this is a common and pragmatic approach, it is critical to acknowledge that this endpoint is highly susceptible to ascertainment bias when comparing pregnant and non-pregnant populations. The reason is that the threshold for hospital admission is likely different between these two groups. Pregnant individuals are, rightly, monitored more closely by the healthcare system. Both patients and clinicians may have a lower threshold for seeking care or recommending hospitalization for respiratory symptoms, out of an abundance of caution for both the parent and the fetus. In contrast, a non-pregnant person of the same age with identical symptoms might be more likely to be managed as an outpatient.
Your analysis of outcomes stratified by COVID-19 severity (lines 334-337) is subject to reverse causality. This bias makes it difficult to interpret the finding that 'severe' COVID-19 is associated with adverse maternal and neonatal outcomes. The issue stems directly from your definition of severe COVID-19 as any hospitalization with a positive COVID-19 test. Consider this highly plausible clinical scenario: a pregnant patient presents with an obstetric emergency, such as threatened pre-term labor or pre-eclampsia, which necessitates her admission to the hospital. Upon routine screening at the hospital, she tests positive for COVID-19, perhaps even asymptomatically. Within your dataset, this patient is now classified as having the 'exposure' (severe COVID-19, because she was hospitalized) and the 'outcome' (the obstetric complication).
In your Discussion section, you must add a paragraph explicitly discussing ascertainment bias as a major limitation. Explain why it occurs (differential surveillance) and how it likely inflates the prevalence of "severe" cases in your pregnant cohort. The conclusion that pregnant people were "7 times more likely to have severe infections" should be rephrased to be more cautious.
Specific Comments
- The tables in the article are quite dense, with information grouped across multiple columns, making quick reading and comparison difficult. I strongly recommend replacing these large tables with more intuitive visualizations to highlight the key results of your investigation better. Tables 1 and 3 (Baseline Characteristics) should be condensed for the main manuscript, with detailed variables moved to a supplementary appendix. You may consider to present a Standardized Mean Difference plot to provide a clear, visual summary of the comparability between your cohorts. Table 4, the most important table in your paper, is the ideal candidate for a Forest Plot.
